

# Migration as a Hidden Risk Factor in Seismic Fatality: A Spatial Modeling Approach to the Chi-Chi Earthquake and Suburban Syndrome

Tzu-Hsin Karen Chen[1,2], Kuan-Hui Elaine Lin[3], Thung-Hong Lin[4], Gee-Yu Liu[5], Chin-Hsun Yeh[5]

, and Diana Maria Ceballos[2]

[1]Department of Urban Design and Planning, University of Washington, Seattle, WA 98105, USA
[2]Department of Environmental and Occupational Health Sciences, University of Washington, Seattle, WA 98105, USA
[3]Graduate Institute of Sustainability Management and Environmental Education, National Taiwan Normal University, Taipei 10645, Taiwan
[4]Institute of Sociology, Academia Sinica, Taipei 11529, Taiwan
[5]National Center for Research on Earthquake Engineering, Taipei 10668, Taiwan

*Correspondence to*: Tzu-Hsin Karen Chen (kthchen@uw.edu) and Kuan-Hui Elaine Lin (khelin@ntnu.edu.tw)

**Abstract.** Suburban areas have disproportionately experienced higher fatalities during major earthquakes. Place-based models attribute this spatial disparity to hazard, exposure, and social vulnerability factors. However, the impact of migration on seismic fatality remains underexplored, primarily due to the challenges in accessing mobility data. In this study, we apply a geospatial method, the radiation model, to estimate migration patterns as a critical component of exposure and vulnerability. Analyzing the 1999 Chi-Chi earthquake in Taiwan with Poisson regression across 4,052 neighborhoods, we factor in migration inflow 20 (i.e., population traveling from other neighborhoods), migrants' origin income, and indigenous population percentage among migrants, along with other risk factors proven in previous studies. Our findings indicate that migration inflow significantly correlates with increased fatalities. Furthermore, a lower income at the migrants' origin neighborhood is significantly associated with higher fatalities at their destination. An elevated proportion of indigenous population in the migrants' origin neighborhood also significantly correlates with increased fatalities, although the impact of the Chi-Chi earthquake does not 25 predominantly affect indigenous jurisdictions. This study underscores the seismic fatality risk in the outskirts of megacities, where migrants from lower income and historically marginalized groups are more likely to reside for precarious employment conditions, emphasizing the need for affordable and safe living infrastructures for the migrating population. Addressing migrants' vulnerabilities in housing will not only reduce seismic fatality risk but also improve preparedness against other disasters and public health emergencies.





## 1 Introduction


Globally, from 1996 to 2015, earthquakes resulted in over 750,000 deaths, comprising 55.6% of all natural disaster fatalities in this period (UNISDR, 2016). Seismic fatalities, while intuitively linked to earthquake exposure, do not always align with population density or hazard magnitude. For example, during Taiwan's 1999 Chi-Chi earthquake, most fatalities occurred not in densely populated urban centers or remote rural areas but in suburban or urban fringe areas around the Taichung metropolitan

region (Fig. 1). Similar mortality patterns were observed in subsequent significant earthquakes in Wenchuan, China (2008), Central Chile (2010), and Gorkha, Nepal (2015), predominantly affecting suburbs or neighborhoods of small-to-medium-sized cities (Adhikari et al., 2021; Allan et al., 2013; Panday et al., 2021; Xu et al., 2009). This pattern, which we term as the 'suburban syndrome,' suggests that the highest fatality rates may be more related to urban development and housing safety issues than to the earthquake's magnitude alone. Nevertheless, this socio-spatial fatality pattern in suburban belts has been

underexplored in seismic risk research.

By 2030, small and middle-sized cities and towns with less than one million population size are set to register 54% of the world's urban population and have undergone rapid growth and urban expansion in recent decades (UN, 2018). Small and middle-sized cities at the outskirts of a metropolitan area often offer more affordable housing for migrant workers as well as lower-income groups who were pulled into cities for job opportunities but displaced from city centers (Taubenböck et al.,

2018). Moreover, many of these small and middle-sized cities feature unregulated development issues such as informal settlements, crowded living conditions, unregulated construction, poor political governance, and limited finance (Birkmann et al., 2016). Specifically, a large proportion of socioeconomically marginalized people such as migrants from rural villages and indigenous people are more likely to reside in these areas for rental housing with a lack of safe structure or maintenance (Andersen et al., 2018; Belanger et al., 2013; Shier et al., 2015). Also, rural population migrate to urban areas for temporary

laboring jobs (e.g., construction) when crops are off season (Fan and Li, 2020; Kumar and Sati, 2023). Urban peripheries that feature laboring jobs and offer more affordable housing are usually more attractive to these migrant workers than downtowns (Ren, 2021). All these conditions have intensified these populations' vulnerability to earthquakes.



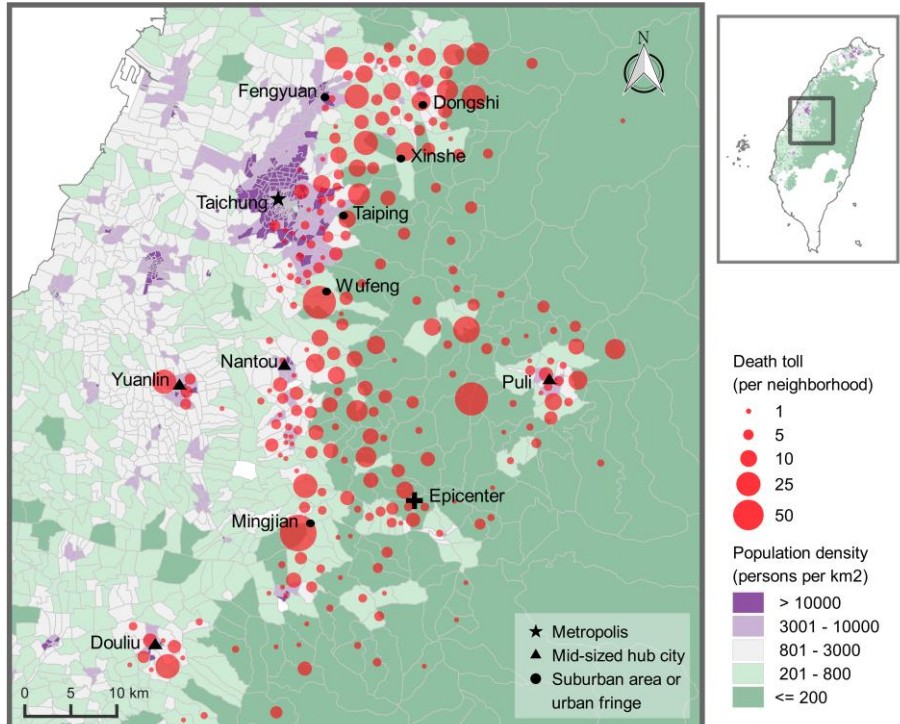

**Figure 1: A case of the suburban syndrome in seismic fatalities. Spatial distributions of fatalities and population density indicate that the neighborhoods in medium-sized hub cities, suburban areas, and urban fringes suffered the most fatalities during the Chi-Chi earthquake. The population data is from Taiwan Census 2000 and fatality data from Tien et al. (2002).**

Seismic risk assessment has primarily focused on individuals' places of residence to estimate exposure and other risk factors (Aldrich and Sawada, 2015; Derakhshan et al., 2020; Lin et al., 2015; Masi et al., 2021; Pavić et al., 2020; Rashed and Weeks, 2003). These measures were able to identify areas with higher exposure and social vulnerability associated with earthquake-related fatalities, thereby informing resource allocation and policy interventions. However, the residence-based approaches overlook the effect of migration patterns at daily, seasonal, and long-term scales that can have a strong impact to reshape risks. More critically, the migration data hardly is reflected in any official population registers. We argue that migration patterns are a vital yet overlooked component in understanding the suburban syndrome in disaster risk assessment. Ignoring population flow between places can lead to an underestimation of exposure, root cause, and progression of vulnerability to environmental hazards (Kumar and Sati, 2023; Wang et al., 2022). For example, seasonal farm workers in North Carolina, USA, faced greater impacts from Hurricane Katrina due to inadequate housing in migrant worker camps (Montz et al., 2011). In Lisbon, Portugal, more than 50% of the population is found in different census locations during the day compared to their residential records, resulting in a 22% higher exposure to the 2012 Lisbon earthquake (Freire and Aubrecht, 2012). Between 1990 and 2010, the migration of the working-age population from rural to urban areas in China led to a 34% increase in the population exposed to seismically hazardous areas, higher than the average population growth at 18% (He et al., 2016). In the United States, exposure to toxic sites is 10% higher when based on people's mobile signal locations than estimates based on their residence (Liu et al.,



2023). These examples underscore the critical role of migration patterns in hazard exposure, necessitating a paradigm shift towards mobility-based risk assessments.

In this study, we hypothesize that incorporating the migration effect into risk models could explain the suburban syndrome, characterized by higher fatalities in small and medium-sized cities and suburban areas. We also hypothesize that migrants from historically marginalized groups, especially indigenous people in colonial societies, may expose to higher seismic risks. Mobility data from recent technological advancements, such as social media records and mobile phone signals, were not widely available before the 2000s. Thus, we introduce the application of the radiation model (Simini et al., 2012) to estimate migration effects on seismic risks. The effect is examined through a case study of the Chi-Chi earthquake that struck Taiwan in 1999. The database for this earthquake encompasses the location of each death and high-resolution ground motion data from 650 stations. By combining earthquake hazard, socioeconomic, indigenous population data, and migration estimates at the neighborhood level, we construct three nested multivariate models. Our analyses address three questions: (i) What are the determinant factors constituting seismic risk in terms of hazard, exposure, and vulnerability? (ii) What roles do migration patterns play in seismic fatalities with respect to the size of the migrant population and the economic status of their origin? (iii) Are historically marginalized populations, especially indigenous people, more vulnerable during migration in terms of seismic risk?

## 1.1 Background for place-based and mobility-based effects in the "suburban syndrome"

When performing seismic risk assessment, studies usually conceive risk factors of a given place by measuring three distinctive components that determine fatality—hazard, exposure, and vulnerability (Bilham and Gaur, 2013; Lin et al., 2015). Hazard refers to natural or human-induced physical events that may have adverse effects on exposed elements, such as populations or buildings. Exposure refers to an inventory of elements in an area in which hazard events may occur. In seismic risk assessments, while 'vulnerability' sometimes refers to the fragility of buildings, we specifically refer this term to social vulnerability, which reflects the sociodemographic context of a neighborhood that is prone to suffering seismic fatalities (IPCC, 2012; Turner et al., 2003). Following assumption that living in a poor neighborhood affects a wide range of individual outcomes (Wilson, 2012), the place-based effect refers to internal physical and socioeconomic context of a neighborhood in which hazard, exposure, and vulnerability interact to shape its risk profile. In addition, the mobility-based effect, as we define here, refers to interactions of the outer linkages of a neighborhood with its migrants from other neighborhoods that affect the risk profile, but yet their mobility has been underestimated. These two mechanisms increase the seismic risk in developing and densely populated neighborhoods along the expanding suburban belt—suburban areas, the urban fringe, and local hub cities and towns.

The place-based effect appears to be magnified when hazard, exposure, and vulnerability are considered to reinforce each other, and fatalities are higher in certain neighborhoods. However, the investigations for these phenomena remain insufficient. Seismic risk studies most frequently report the concentration of disadvantaged or vulnerable populations living in poorly constructed buildings (e.g., low-rent or low-price residential and commercial property or slums) in seismic-hazard-prone areas.



During the past four decades, construction booms especially have been witnessed in cities worldwide, resulting in an unprecedented increase of building stocks constructed using inferior materials and assembly methods (Bilham and Gaur, 2013). Most such buildings occur in developing countries or regions where building standards were poorly regulated or enforced. It is deemed that earthquake-resistant buildings are often exclusive to the economically developed neighborhoods. The expensive real estate with high-standard building codes are usually not affordable for the socially disadvantaged population (i.e., low-

income, unemployed, and minority people; transient-migrant or commuting workers; and people who are less educated) in the earthquake zones (Önder et al., 2004). Overall, corruption in the construction industry, absence of education on earthquakes, prevalence of poverty, and income inequality, which relates to segregation in earthquake zones, are responsible for fatalities associated with building collapses (Anbarci et al., 2005; Escaleras et al., 2007). There are still other factors that can heighten social and geographical segregation during earthquakes. Public or social housing is often built on newly derived lands that are

prone to various types of hazard threats, such as those close to transportation, hillsides, riverbanks, or coasts, under city planning (Cutter et al., 2006). Proximity to industrial sites during an earthquake can increase the risk of explosions and fires, potentially leading to fatalities (Moghaddam et al., 2023). The spatial segregation of building quality and unequal exposure to hazards can be worsened under rapid and unregulated urban expansion (Brouwer et al., 2007; Lavell, 2003; McGranahan et al., 2007). Therefore, during building booms particularly, the rapidly developing yet relatively poor neighborhoods around the

suburban belt could therefore face greater exposure to hazards with a higher proportion of buildings with low-quality codes to enhance seismic risks.

Another mechanism, the mobility-based effect, which focus on human mobility across neighborhood boundaries, could profoundly influence earthquake fatalities through migration behavior. Human mobility shapes population dynamics at

different timescales from a day to several seasons in a year. The simplest view of human mobility is the "push–pull" theory, which states that a pull-in force occurs in places with more favorable conditions and opportunities that attract migration, whereas a push-out force occurs in places with fewer opportunities and more constrains that force people to move out of those places (Ravenstein, 1889). These processes cause short-term population flow (e.g., commuting daily to work, weeks/months in temporary jobs, and semesters/years at school) and long-term migration; these population flows can take place in both formal

and informal forms. Research indicates that city scale (population or accumulated goods) and job opportunities are critical factors of a pull process (Jamshed et al., 2020; Simini et al., 2012). The gravity model, for example, uses two elements— population number and distance between places—to predict population flows (Simini et al., 2012; Viboud et al., 2006). Survey studies also find that low family income (Fell et al., 2004), unemployment (Böheim and Taylor, 2002), and the cost of commuting are key factors in a push process. All of these dynamisms form the prototype of the relationships in urban systems

and urban–rural linkages (Dicken et al., 2001; Jessop et al., 2008). Ignoring such socio-spatial interactivities could lead to underestimation of the population exposure, especially for hub cities with high population mobility (Lall and Deichmann, 2012). In this regard, the mobility-based effect boosts seismic risk in suburbs and medium-sized hub cities. Sharp and Clark (2008) studied commuting behaviors among urban, suburban, fringe, and rural areas and found that the fringe area is often



marked with a high proportion of new buildings and high mobility. Exposure and resource scarcity (vulnerability) are two
mechanisms occurring simultaneously and linking the suburban and fringe commuting population to intensified risk. Thus, disaster risks are likely to be underestimated in such areas.

## 2 Methods

### 2.1 Taiwan's Chi-Chi earthquake study area

Taiwan is located on the western edge of the Pacific Ring of Fire, a seismically active zone at the convergence of the Philippine
Sea Plate and the Eurasian Plate. Due to complex and active tectonic settings, Taiwan experiences frequent earthquakes. On average, 18,649 earthquakes have hit Taiwan annually from 1991 to 2004, among which 1,047 earthquakes could be sensed (CWB, 2019). The Chi-Chi earthquake was the most lethal earthquake since the 1935 Shinchiku-Taichū earthquake. It struck at 01:47 (GMT+8) on September 21, 1999, in Chi-Chi Township, Nantou County (23.77° N latitude, 120.98° E longitude, Fig. 2), southeast of the Taichung metropolis, the second largest city of Taiwan. The sequence produced a 90 km surface rupture
along the Chelungpu thrust fault stretching between foothills of the Central Mountain Ranges in the east to the alluvial plains and basins in the west (Fig. 2). The MW 7.6 main shock and aftershocks resulted in either the complete destruction or the serious damage of approximately 110,000 buildings and resulted in 2,444 deaths, 94% of which were due to building collapses (Kao and Chen, 2000; Tien et al., 2002). Fatalities were concentrated in Taichung County (1,138 deaths) and Nantou County (928 deaths). Most fatalities occurred in suburbs or urban fringes along the periphery of the Taichung metropolis and medium-
sized cities outside of the metropolis (Fig. 1). Previous studies on the Chi-Chi earthquake found known risk factors from the hazard, exposure, and vulnerability components in risks (Lin et al., 2017; Lin et al., 2015). Yet, migration effects and factors of the disadvantaged population remain unknown.

In this study, we adopted a risk formula (IPCC, 2012; UNDRO, 1980) to estimate the socio-spatial effect of the seismic fatalities in the Chi-Chi earthquake of Taiwan. The smallest (forth-level) administrative division in Taiwan is the village level
in rural districts, which is equivalent to neighborhood in urban districts. As village and neighborhood refer to the same administrative level, we hereafter use neighborhood to harmonize the terminologies and use rural or urban neighborhood for specifying the context as necessary. In total, Taiwan encompassed 7,558 neighborhoods with an average population of 3,000 in 1999. Although the earthquake was sensible island wide, it did not yield tangible damage across the entire island. Thus, we excluded neighborhoods under the threshold of ground motion intensity, Sa03 of 0.14g, which suggests that no damage
occurred below this intensity (Lin et al., 2015). This resulted in 4,502 neighborhoods as our study area.



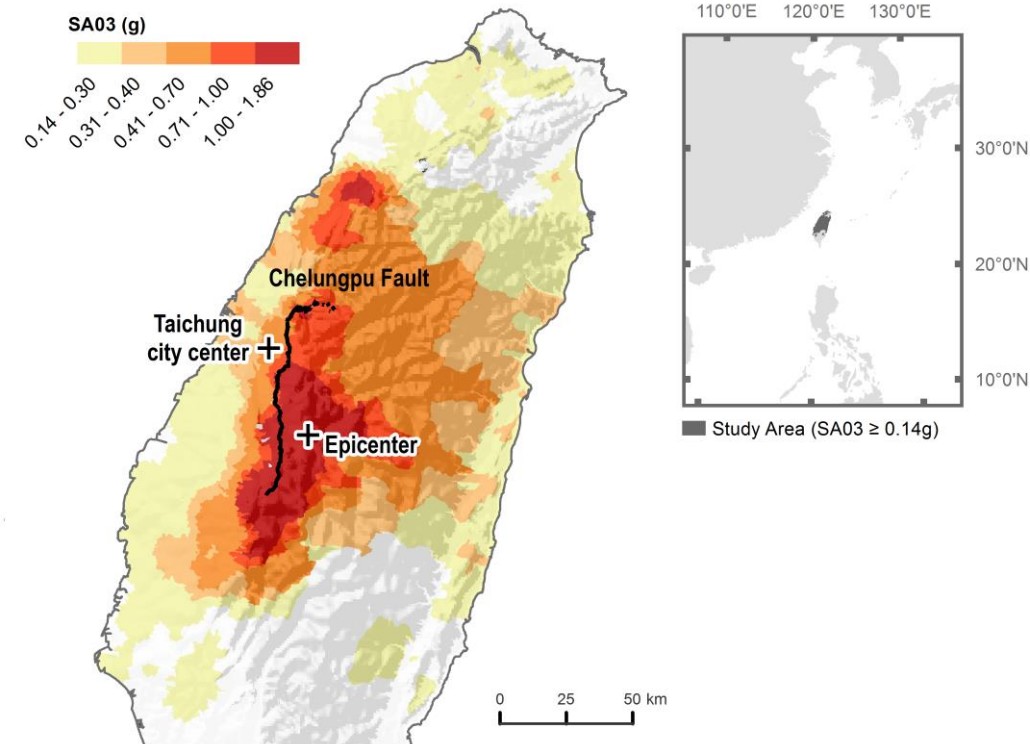

**Figure 2: Map of Taiwan's mainland and the study area (Sa03 ≥ 0.14g). The Chi-Chi earthquake resulted in a 90 km surface rupture along the Chelungpu thrust fault stretching between the foothills of the Central Mountain Range in the east to the alluvial plains and basins in the west. Graphic underlay showing terrain shades is from the ASTER Global Digital Elevation Map (Tachikawa et al., 2011).**

## 2.2 Data

### 2.2.1 Dependent Variable: Earthquake Fatalities

The dependent variable in this study is the death toll in each neighborhood. We obtained the data from Tien et al. (2002) who investigated the fatalities caused by the Chi-Chi earthquake. Notably, the research team from the National Central University tracked the location of each death using the global positioning system (Kao and Chen, 2000; Tien et al., 2002). The death toll thus reflected the location of the death rather than their registered residence. The fatalities per neighborhood ranges from 0 to 87 (Table 1).



**180 Table 1: Descriptive statistics of the selected variables at the neighborhood level.**

| Dependent variable | Mean | SD | Min. – Max. | Source |
|---|---|---|---|---|
| Fatalities | 0.52 | 3.79 | 0–87 | Tien et al. (2002) |
| | | | | |
| **Independent variable** | | | | |
| *Hazard* | | | | |
| Sa03 (g) | 0.36 | 0.27 | 0.14–1.86 | CWB* |
| Fault-impacted area | 0.02 | 0.13 | 0–1 | Chen et al. (2001) |
| *Exposure* | | | | |
| Population (unit: 10,000 people) | 0.29 | 0.26 | 0–3.88 | Taiwan Census 2000 |
| Percentage of low seismic capacity building | 0.37 | 0.21 | 0–1 | NCREE †(Yeh et al., 2006) |
| *Vulnerability* | | | | |
| Sex ratio | 1.08 | 0.31 | 0.19–11.86 | Taiwan Census 2000 |
| Percentage of population under the age of 15 | 0.2 | 0.04 | 0–0.41 | Taiwan Census 2000 |
| Percentage of population over the age of 64 | 0.11 | 0.05 | 0–0.65 | |
| Ln (Median household income) | 6.24 | 0.17 | 5.41–7.26 | Ministry of Finance |
| Ln (Standard deviation of household income) | 6.31 | 0.5 | 5.2–10.07 | |
| Indigenous population proportion | 0.03 | 0.14 | 0–0.98 | Ministry of Interior |
| *Migration pattern* | | | | |
| Ln (Estimated migration inflow) | 0.78 | 0.93 | -3.69–3.86 | Radiation model (this study) |
| Ln (Average income of migrants' origin) | 6.25 | 0.15 | 5.76–6.98 | Radiation model (this study) |
| Indigenous population proportion in migrants | 0.04 | 0.13 | 0–0.93 | Radiation model (this study) |

\* Central Weather Bureau
† National Center for Research on Earthquake Engineering

### 2.2.2 Independent Variable: Hazard

The seismic hazard dimension comprises two variables: seismic intensity and surface rupture. For seismic intensity, we measured Sa03, standing for spectral acceleration at 0.3 second. This measurement is found more representative than peak ground acceleration or peak ground velocity when considering building damages (Lin et al., 2015; Wu et al., 2004; Wu et al., 2002). The Sa03 ground motion data was obtained from the Central Weather Bureau (CWB), which had installed approximately 650 free-field strong motion stations around Taiwan before the Chi-Chi earthquake under the Taiwan Strong

Motion Instrumentation Program (Shin et al., 2003). In terms of surface rupture, the Chelungpu Fault surveyed by Chen et al. (2001) was applied to identify the distribution of the fault rupture. Chelungpu is a thrust fault; empirically major destruction



occurred on the hanging wall side of depression. We thus identified a fault-impact zone with a 5-km buffer (4 km on the hanging wall and 1 km on the footwall) along the fault. The ratio of the area in the fault-impact zone (from 0 to 1) in each neighborhood was calculated based on the geographical information system to reflect its influence on fatalities.

### 2.2.3 Independent Variable: Exposure

The exposure variables included population size and proportion of buildings with low-seismic capacity (Lin et al., 2017; Lin et al., 2015). We obtained the population data from the Taiwan Population and Household Census 2000 (Chang and Shyue, 2009). The average population per neighborhood was 2,901, with a minimum of 16 people in a rural village and a maximum of 38,822 in the densest neighborhood of the urban areas. Additionally, building fragility is a crucial intermediate factor that can fundamentally influence earthquake fatalities (Bilham and Gaur, 2013; Birkmann et al., 2016; Lin et al., 2015; Yeh et al., 2006). People situated in fragile buildings with low seismic resistance would suffer from a higher degree of exposure from the doubling effect of the initial seismic hazard and the potential collapse of buildings. We obtained the building seismic capacity data from the National Center for Research on Earthquake Engineering (Yeh et al., 2006). The building capacity classification was based on the history of seismic design codes for the buildings in four levels (high, moderate, low and pre-code) (Scawthorn et al., 2006). We considered pre-code and low-code buildings as low seismic capacity and calculated the percentage of the floor area with low seismic capacity per neighborhood in the year 2000. The average percentage of low seismic capacity buildings was 37% among the studied neighborhoods (Table 1).

### 2.2.4 Independent Variable: Vulnerability

The vulnerability variables included neighborhood-level sex ratio (male/female population), percentage of population under the age of 15, percentage of population over the age of 64, household income, income inequality, and indigenous population proportion. Sex ratio, dependent population (age under 15 and over 64), median income, and income inequality are known risk factors for Chi-Chi earthquake death tolls (Li et al., 2017; Lin et al., 2015). The population dependency factor calculates the percentage of the population under the age of 15 and over the age of 64. A larger dependent population, either young or aged, indicates a higher degree of vulnerability. Household income, measured by median and standard deviation, reflects the average economic development and income inequities of each neighborhood, respectively, which are key factors of vulnerability (Cutter et al., 2006; Cutter and Finch, 2008; Kahn, 2005).

We integrated the proportion of the indigenous population as a new variable to reflect the land dispossession and economic marginalization that occurred since the colonial era. The Ministry of Interior in Taiwan defines indigenous populations as groups of people with their own distinct languages, cultures, and traditions, and acknowledges the existence of 16 official indigenous groups. As of 2024, their population numbered 589,038, representing approximately 2.5% of Taiwan's total population of 23 million (Ministry of Interior, 2024). In contrast to the majority Han population, who migrated to Taiwan in different stages in the last centuries, the indigenous peoples belong to the Austronesian language family and have inhabited





the island for millennia. The colonial history of Taiwan began in 1624 with the Dutch Republic (Dutch East India Company),
followed by Spain, China (during the Ming and Qing dynasties), Japan (1895-1945), and lately the Chinese Nationalist Party
(also known as Kuomintang, KMT), who settled in Taiwan after world war II and lifted martial law in 1987 (Nesterova and
Jackson, 2018). Over the political transitions, the indigenous peoples have lost their lands, with most remaining indigenous
territories concentrating in mountainous regions. Hence, agriculture has been the primary employment sector for indigenous
peoples, comprising 20% of their labor force, compared to 6.6% of agricultural employment in the total workforce in 2002 in
Taiwan (Council of Indigenous Peoples, 2005). Manufacturing (15%) and construction (13%) are the next largest sectors for
indigenous employment. Nevertheless, indigenous occupations often fall into the precarious category known as "3K jobs." The
term "3K jobs" derives from Japanese words meaning "hard (kitsui)," "dirty (kitanai)," and "dangerous (kiken)," respectively.
Typical 3K jobs are found in manufacturing and construction, usually offered on a contract basis with daily wages two to three
times higher than the minimum wage, making them an attractive option for farmers during the off-season. It is worth noting
that 64% of indigenous farmers experience off-seasons ranging from one to six months, leading them to seek temporary work
and housing in nearby towns (Council of Indigenous Peoples, 2005).

We obtained the household income data before tax in 1999 from the Ministry of Finance (2023). Notably, the median annual
household income of each neighborhood ranges from 224,000 TWD (~7,400 USD) to 1,425,000 TWD (~47,500 USD), and
the standard deviation of annual household income ranges from 182,060 TWD (~6,068 USD) to 23,734,840 TWD (~791,161
USD); these values indicated considerable social differences between neighborhoods. We used the logarithmic transformation
of the median and standard deviation of household income to deal with the skewed distributions of these variables. We obtained
demographic data at the neighborhood level from Taiwan Population and Household Census 2000. The overall sex ratio was
1.08 on average, meaning that males outnumber females in the studied neighborhoods. There were no statistics available
regarding the non-binary gender population. We obtained the proportion of indigenous population data from the Ministry of
Interior (2023). The available neighborhood-level data closest to our study year was documented in 2008, and we considered
the indigenous population distribution between 1999 and 2008 to be similar.

**2.2.5 Independent Variable: Migration Pattern**

In Taiwan, during the 1999 earthquake, no statistics were available on internal migration at the neighborhood level. Therefore,
we utilized a radiation model to estimate population migration at this granularity (Simini et al., 2012). This model, which is
widely used to determine migration flows between origin and destination areas, utilizes population data and pairwise distances
between neighborhoods to represent the push-and-pull factors in migration (Gibb et al., 2023; Li et al., 2017; Simini et al.,
2012; Yang et al., 2014). The formula for the model uses the product of the populations of the origin and destination
neighborhoods as the numerator. The denominator is an adjusted product of the populations within the vicinity of both the
origin and destination areas, represents the effect of competing neighborhoods on the population flow. We applied the radiation
model to quantify $T_{ij}$, the proportion of the migrants from neighborhood $i$ moving to neighborhood $j$, as follows in Eq. 1:





$$T_{ij} = \frac{Pop_i \, Pop_j}{(Pop_i + S_{ij})(Pop_i + Pop_j + S_{ij})}$$  (Eq. 1)

where $Pop_i$ and $Pop_j$ are the populations of neighborhoods $i$ and $j$, respectively; $S_{ij}$ is the total population within the radius $r_{ij}$, centered at $i$ (excluding the source and destination populations); $r_{ij}$ is the distance between the centroids of neighborhoods $i$

and $j$. Based on this origin-destination matrix, we produced three variables: estimated migration inflow, average income of migrants' origin, and the proportion of indigenous population in migrants.

Estimated migration inflow for neighborhood $j$ can be represented as $\sum_i T_i T_{ij}$, where $T_i$ is the total number of migrants originating from $i$. Although $T_i$ is an unknown parameter, it is proportional to $Pop_i$ based on Simini et al. (2012). Therefore,

we calculated the estimated migration inflow for neighborhood $j$ as $\sum_i Pop_i T_{ij}$, which is a relative size proportional to the total number of migrants in a neighborhood $j$. Specifically, the estimated value represents the size of migrant inflow assuming that every place has 100% of its population migrating. In reality, however, the inflow is smaller and proportional to this estimate (Simini et al. 2012). Since this variable does not represent the absolute number of migrants but rather their relative size, we used the logarithm of the relative size of migrant inflow to interpret the effect of a percent change in migrant size on the percent

change in fatalities.

We calculated the average income of migrants' origin by weighting the origin's median income by the relative migrant inflow among all migrants in neighborhood $j$ (Eq. 2). Similar to the neighborhood's income variable, we used the logarithm transformation of the variable to deal with its skewed distribution.

$$\frac{\sum_i Pop_i T_{ij} \times income_i}{\sum_i Pop_i T_{ij}}$$  (Eq. 2)


We calculated the proportion of indigenous population in migrants based on the proportion of indigenous people and the relative migrant inflow. Assuming the proportion of migrants being indigenous from neighborhood $i$ equals to the proportion of indigenous population in neighborhood $i$, the proportion of indigenous population in the migrants to neighborhood $j$ was calculated as Eq. 3.

$$\frac{\sum_i Pop_i T_{ij} \times indigenous\%_i}{\sum_i Pop_i T_{ij}}.$$  (Eq. 3)


We used the library 'spdep' in R to calculate distance between neighborhoods (see the Code Availability section).



**2.3 Models**

We used Poisson regression and maximum likelihood estimation to predict incidence rate ratio and significance levels for each
covariate from the risk components of hazard, exposure, and vulnerability. The incidence rate ratio of a given variable can be interpreted as the factor by which the fatality rate multiplies when that variable increases by one unit, assuming all other variables remain constant (Clayton and Hills, 2013). We modeled neighborhood-level fatalities against these variables, assuming a linear response association, across 4,052 neighborhoods in Taiwan. We started with known risk factors in the basic model (Model 1; Eq. 4). Subsequently, we incorporated the proportion of the indigenous population in the model (Model 2;
Eq. 5). Lastly, we included estimated characteristics of migration patterns (i.e., migrant inflow, average income of migrants' origin, and indigenous population proportion in migrants) in the final model (Model 3; Eq. 6). The three nested regression models can be described as follows:

$$\log E(\text{FAT}_i/x) = \beta_0 + \beta_1 \text{Sa03}_i + \beta_2 \text{Fault}_i + \beta_3 \text{Pop}_i + \beta_4 \text{LowCapacityBuilding}_i + \beta_5 \text{SexRatio}_i + \beta_6 \text{AgeUnder15}_i +$$
$$\beta_7 \text{AgeOver64}_i + \beta_8 \text{MedianIncome}_i + \beta_9 \text{StdIncome}_i + \varepsilon_i \qquad \text{(Model 1, Eq. 4)}$$


$$\log E(\text{FAT}_i/x) = \beta_0 + \beta_1 \text{Sa03}_i + \beta_2 \text{Fault}_i + \beta_3 \text{Pop}_i + \beta_4 \text{LowCapacityBuilding}_i + \beta_5 \text{SexRatio}_i + \beta_6 \text{AgeUnder15}_i +$$
$$\beta_7 \text{AgeOver64}_i + \beta_8 \text{MedianIncome}_i + \beta_9 \text{StdIncome}_i + \beta_{10} \text{Indigenous}_i + \varepsilon_i \quad \text{(Model 2, Eq. 5)}$$

$$\log E(\text{FAT}_i/x) = \beta_0 + \beta_1 \text{Sa03}_i + \beta_2 \text{Fault}_i + \beta_3 \text{Pop}_i + \beta_4 \text{LowCapacityBuilding}_i + \beta_5 \text{SexRatio}_i + \beta_6 \text{AgeUnder15}_i +$$
$$\beta_7 \text{AgeOver64}_i + \beta_8 \text{MedianIncome}_i + \beta_9 \text{StdIncome}_i + \beta_{10} \text{Indigenous}_i + \beta_{11} \text{MigrationInflow}_i +$$
$$\beta_{12} \text{MigrantIncome}_i + \beta_{13} \text{MigrantIndigenous}_i + \varepsilon_i \qquad \text{(Model 3, Eq. 6)}$$

The response $FAT_i$ refers to the fatality count in neighborhood $i$. For each neighborhood $i$, hazard variables included $Fault_i$ (proportion of the area in the fault zone) and $SA03_i$ (the ground motion intensity measured as Sa03 (g)). Exposure variables
included $POP_i$ (population) and $\text{LowCapacityBuilding}_i$ (proportion of buildings with low capacity). Vulnerability variables included $\text{SexRatio}_i$ (ratio of males to females), $\text{AgeUnder15}_i$ (percentage of the population under age 15), $\text{AgeOver64}_i$ (percentage of the population aged 65 or older), $MedianIncome_i$ (the logarithm of the median household income in neighborhood $i$), $StdIncome_i$ (the logarithm of the standard deviation of household income in neighborhood $i$), and $\text{Indigenous}_i$ (proportion of indigenous people in neighborhood $i$'s population). Migration pattern variables, estimated by the
radiation model, included $\text{MigrationInflow}_i$ (the logarithm of relative size of migrant inflow to neighborhood $i$), $\text{MigrantIncome}_i$ (the logarithm of average income of the migrants' origin in neighborhood $i$), and $\text{MigrantIndigenous}_i$ (percentage of indigenous people among migrants in neighborhood $i$). The parameter $\varepsilon_i$ represents the error term. We assessed model performance by using $R^2$ and log-likelihood values. We performed all statistical computations in R (version 4.3.1); statistical significance was evaluated at $p<0.05$.





## 3 Results

### 3.1 Estimated migration patterns

Using the radiation model, we characterize the relative size of migration inflow, average income of migrants' origin, and the percentage of indigenous people among migrants (Fig. 3). Two example neighborhoods show they attracted migrants from different areas and distances based on their geography. A neighborhood in Wufeng (Fig. 1), a regional hub in the outskirts of Taichung metropolitan area, was estimated to have migrants from other neighborhoods in around 10 km, as well as some migrants from Nantou, a mountain city about 40 km away. In contrast, the neighborhood in Puli (Fig. 1), a mountainous town, was estimated to have migrants from other mountainous villages that are up to 80 km away. This shows that the radiation model captures long-distance migration patterns between remote villages and their closest large town.

Comparing the patterns of migrants and residents, higher migration inflow, income of origin and percent indigenous population among migrants were aligned with the spatial pattern of population, population size, and percent indigenous population among residents. Yet, there is some differences between the patterns of migrants and residents. A higher migration inflow was estimated in the suburbs and a lower migration inflow in rural areas and city centers, as city centers have more competing destinations along the way from the rural area to the city. The radiation model also highlights a distinct zone of indigenous migrants at the foothills along the indigenous neighborhoods in mountainous areas (Fig. 3). This observation supports various case studies that highlight indigenous migrants' preferences for blue-collar jobs in towns and cities, while still staying close to their homeland.



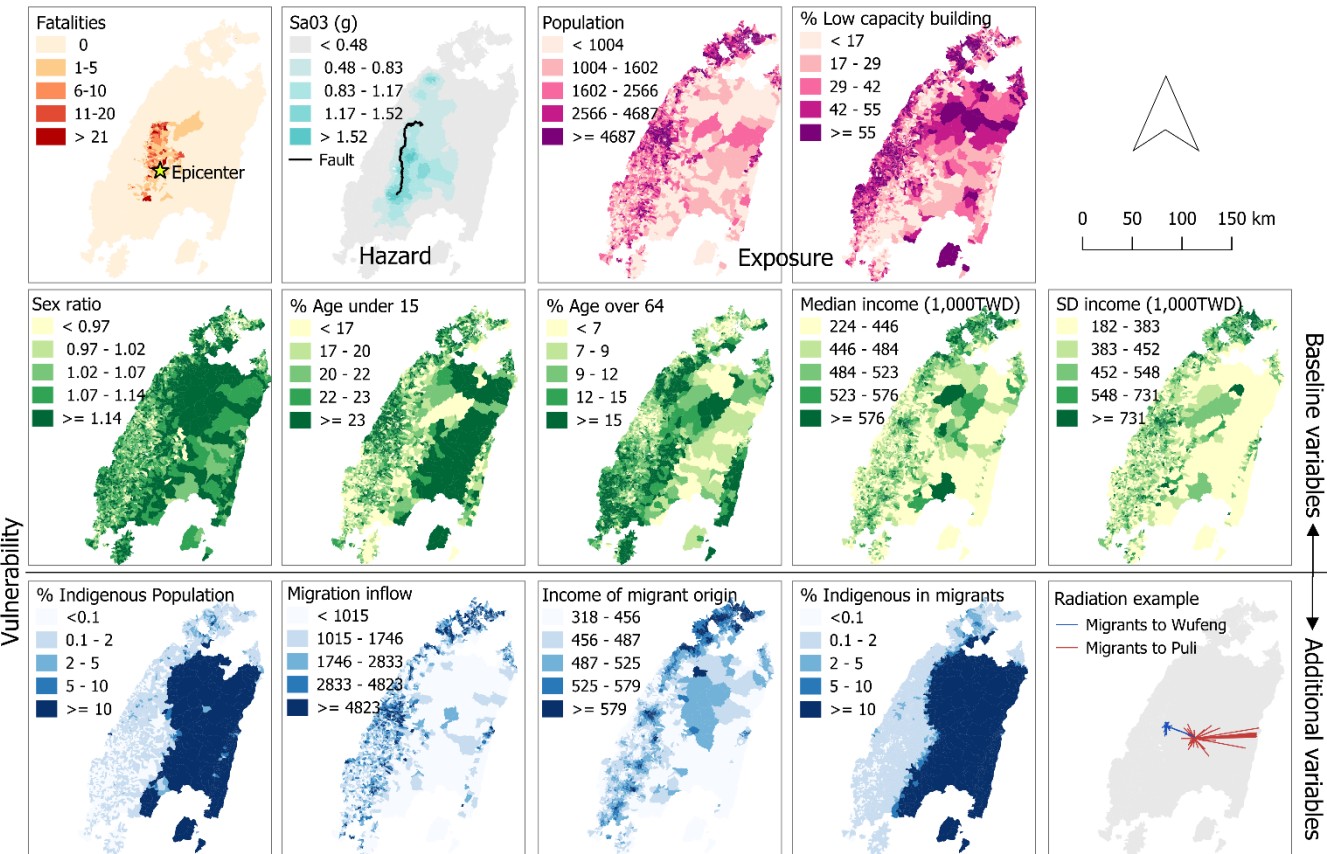

**Figure 3: Maps of seismic fatality risk factors, including hazard, exposure, vulnerability, and migration pattern variables. Migrant inflow, as well as income and the proportion of the indigenous population within the migrant inflow, are estimated using the radiation model. Examples from migrant origins in Wufeng and Puli demonstrate that towns in remote areas (e.g., Puli) act as job opportunity hubs for villages from a long distance.**

## 3.2 Risk factors of seismic fatalities

The 1999 Chi-Chi earthquake resulted in 2,444 deaths. Model 1 confirms the findings of Lin et al. (2015), indicating that seismic intensity, fault impact zone, population, buildings with low seismic capacity, female ratio, lower household income, and higher income inequality are associated with increased fatalities (Table 2). Seismic hazard remains determining factors influencing the distribution of fatalities in the Chi-Chi earthquake. A 0.1 unit increase in Sa03 could lead to a 173% rise in neighborhood fatalities (mean Sa03 = 0.36), assuming other variables remain constant. The presence of a fault zone covering 10% of a neighborhood's area resulted in a 90% increase in fatalities. Due to the correlation between seismic intensity and proximity to the fault, the interplay between these two variables is a primary determinant of fatalities. Exposure variables also show significant associations with fatalities. We find that an increase of 2,000 residents in a neighborhood could result in a



75% rise in fatalities, while a 20% increase in the percentage of buildings with low seismic capacity could result in a 20% rise in fatalities. We also find that several vulnerability variables are significant. Each 10% increase in a neighborhood's male-to-female ratio is associated with a 3.5% decrease in fatalities. Each 0.2 unit increase in Ln (median household income), for
example, from 403,000 TWD (~12,587 USD) to 493,000 TWD (~15,400 USD), could reduce fatalities by 20%. Each 0.5 unit increase in Ln (standard deviation of household income), for example, from 403,000 TWD (~12,587 USD) to 665,000 TWD (~20,800 USD), could lead to a 40% increase in fatalities. The only variable presenting a different direction of association compared to Lin et al. (2015) is the percentage of population aged over 64. Our findings suggest that a 10% increase in a neighborhood's population over age 64 is associated with 10 % lower fatalities. This discrepancy is likely due to our use of
higher resolution data at the neighborhood level, as opposed to the previous study's district-level analysis, and the observation that higher death tolls occurred in neighborhoods predominantly aged 15-64.

Model 2, which includes the proportion of the indigenous population in addition to Model 1's variables, indicates a negative association of the indigenous population with fatalities (Table 2). It suggests that every 10% increase in a neighborhood's indigenous population is associated with a 7.5% reduction in fatalities, demonstrating that the impact of the Chi-Chi earthquake
is not primarily concentrated in indigenous jurisdictions. However, Model 3, which incorporates migration variables, reveals that a higher proportion of indigenous people among the migrants is linked to increased fatalities. Specifically, a 10% increase in the indigenous proportion among migrants correlates with a 20% increase in fatalities. Model 3 also shows that higher migration inflow and a lower average income in migrants' places of origin are both associated with increased fatalities (P < 0.05).




**Table 2: Incidence rate ratios and standard errors estimated by Poisson models for neighborhood-level risk factors, estimated by three nested models. Model 1 includes hazard, exposure, and vulnerability variables previously found in Lin et al. (2015), Model 2 adds the indigenous population proportion, and Model 3 includes migration pattern variables.**

| Neighborhood-level covariates | Incidence Rate Ratio (Standard Error) | | |
|---|---|---|---|
| | Model 1 | Model 2 | Model 3 |
| *Hazard* | | | |
| Sa03 (g) | 27.26 (1.07)*** | 26.72 (1.07)*** | 21.44 (1.07)*** |
| Fault-impacted area | 10.06 (1.06)*** | 9.82 (1.06)*** | 10.17 (1.06)*** |
| | | | |
| *Exposure* | | | |
| Population (unit: 10,000 people) | 4.78 (1.06)*** | 4.74 (1.06)*** | 4.46 (1.06)*** |
| Percentage of low seismic capacity building | 2.01 (1.13)*** | 1.9 (1.13)*** | 1.37 (1.13)* |
| | | | |
| *Vulnerability* | | | |
| Sex ratio | 0.65 (1.1)*** | 0.65 (1.1)*** | 0.7 (1.11)*** |
| Percentage of population under the age of 15 | 0.28 (2.26) | 0.65 (2.34) | 0.28 (2.38) |
| Percentage of population over the age of 64 | 0 (2.1)*** | 0 (2.13)*** | 0 (2.29)*** |
| Ln (Median household income) | 0.1 (1.2)*** | 0.08 (1.22)*** | 0.4 (1.29)*** |
| Ln (Standard deviation of household income) | 1.79 (1.05)*** | 1.79 (1.05)*** | 1.81 (1.05)*** |
| Indigenous population proportion | | 0.25 (1.4)*** | 0.08 (1.82)*** |
| | | | |
| *Migration pattern* | | | |
| Ln (Estimated migration inflow) | | | 1.24 (1.04)*** |
| Ln (Average income of migrants' origin) | | | 0.02 (1.37)*** |
| Indigenous population proportion in migrants | | | 2.93 (1.57)* |
| | | | |
| N | 4502 | | |
| Log likelihood | -4371 | -4359.6 | -4252.4 |
| Pseudo R-square | 0.49 | 0.492 | 0.504 |

Note: ***P-value < 0.001, **P-value < 0.01, and *P-value < 0.05




## 4 Discussion

This study presents novel models to understand migration as a risk factor of seismic fatalities which has been long underestimated due to lack of mobility data. We find that not only place-based risk factors contribute to fatality risk in the Chi-Chi earthquake. Rather, after adjusting for seismic intensity, population exposure, and residence-based socioeconomic factors,
higher fatality risk was among neighborhoods attracting migrants of lower origin income or/and indigenous migrants. This finding, along with the migration patterns concentrating in the suburban areas outside large cities as well as medium-sized cities serving as regional hubs, help explain the suburban syndrome of seismic fatalities.

Our basic model aligns with previous studies, confirming that hazard variables (fault impact and ground motion intensity),
exposure factors (population and percentage of buildings with low seismic capacity), and socioeconomic vulnerability indicators (median and standard deviation of household income) are all determinants of seismic fatality at the neighborhood level. Consistent to previous studies (Lin et al. 2015), hazard and exposure factors together constitute the most important part (an $R^2 = 0.4$) contributing to seismic fatality. Social vulnerability variables, although play a marginal effect (an increase of $R^2 < 0.1$), are significantly associated with fatality disparities. That is to say, when seismic intensity and exposure are at similar
levels, it is the composition of social vulnerability that could either exacerbate or mitigate the scale of seismic fatality. The basic model results explain the fatality distribution shown in Fig. 1. The highest fatalities appeared in the medium-sized cities in suburbs (e.g., Dali, Taiping, and Fengyuan) and urban fringes (e.g., Wufeng, Dongshi, and Daliao) of the Taichung metropolis. Those are considered as satellite cities and regional hubs in the urban–rural corridor. Remarkably, percent buildings with low seismic capacity are significantly correlated with fatalities and presented a predominant effect explaining high
fatalities in suburbs with lower costs of living. The analysis reflects the view of buildings becoming weapons of mass destruction. Additionally, small and medium-sized cities often suffer from socio-political difficulties of poor governance and insufficient capacity to cope with hazards when experiencing rapid sprawl (Bilham, 2013; Bilham and Gaur, 2013). This calls for urgent seismic risk reduction efforts in suburban neighborhoods and hub cities or towns that could undergo intensified hazard exposure and vulnerability during an earthquake.


Our final model demonstrates that the correlation between migration patterns and fatalities is significant. Although with a $R^2$ only increase 0.014 compared to the base model, the effect is statistically significant. And most importantly, by adding the various migration variables, the roles of migration patterns become even clearer. Not only the estimated migrant inflow is associated with higher fatalities, but the model also shows that lower income at the origin and a higher proportion of indigenous
people among migrants correlate with increased fatalities. Thus, social vulnerability within migrant groups significantly contributes to heightened risks, beyond mere underestimated exposure. Migrant workers often face a myriad of social vulnerabilities (Ahonen et al., 2007; Flynn, 2018), which can exacerbate their risks of death during an earthquake. The vulnerability of migrant workers often leads to a high likelihood of taking risky, non-contractual, temporary employment such



as that in construction or agriculture, often without sufficient benefits or safeguards (Ahonen et al., 2018; Al-Tarawneh et al.,
2020; Foley, 2017; Foley et al., 2014; Howard, 2017). Precariousness in their jobs often provides economic precarity and
results in many immigrant/migrant workers living in overcrowded, substandard housing arrangements, including makeshift
dwellings, trailers, or employer-provided accommodations (Caxaj et al., 2024). These living conditions often lack basic safety
features, such as fire exits or structural stability, increasing their vulnerability to injury and death during an earthquake.

Migrant workers often experience what has been termed as 'overlapping vulnerabilities' (Ceballos et al., 2020; Flynn et al.,
2015) or 'cumulative precarity' (Gravel and Dubé, 2016), which create disparities in health beyond seismic risks (Benach et
al., 2011; Facey and Eakin, 2010; Krieger, 2010). For example, migrant workers experienced higher rates of illness and death
during the COVID-19 pandemic, which is also associated with their commuting behavior and housing conditions (Fielding-
Miller et al., 2020; Istiko et al., 2022; Reid et al., 2021). In the context of the Chi-Chi earthquake in Taiwan, the basic model
reveals a place's general risk profiles. However, the estimation of the actual risk becomes more accurate and precise only after
incorporating migration variables to account for this hidden risk. Overall, indigenous hometowns in the mountainous regions
were less affected by the Chi-Chi earthquake. In contrast, entering small and medium-sized cities and towns for precarious
jobs significantly exposes them to earthquake disasters and social vulnerability. Indigenous populations in Taiwan face an
elevated risk of fatality compared to the general population (Juan et al., 2016) due to various social vulnerabilities, including
low income, limited education, and inadequate access to healthcare (Liao et al., 2024). They are found especially vulnerable
when migrating into cities for job opportunities but only affordable for inferior rental housing conditions that create residential
segregation (Hu and Chen, 2011; Wu et al., 2018). Therefore, our findings suggest that safety regulations on rental housing,
particularly affordable options, as well as rent subsidies for low-income, migrants, and historically marginalized groups, are
imperative strategies to reduce their risk.


As far as we are aware, this is the first study that incorporates migration dimensions into the hazard-exposure-vulnerability
model of seismic fatality risk. Nevertheless, this study is not without its limitations. Given the early timeline of the case study,
we were unable to validate the migration patterns using newer technology. Future studies on recent earthquakes could employ
mobile data to further distinguish between different migration behaviors (e.g., daily commuting versus seasonal migration),
which would more precisely dissect the role of mobility in seismic risks. Additional sociodemographic variables of migrant
workers, such as languages spoken, occupation at the time of the earthquake, and social networks including tribal affiliations,
can further elucidate the sources of vulnerability and resilience to disasters. This study reveals general relevance and opens
scientific questions for future research on migrants' exposure to environmental hazards.

**5 Conclusions**

435       In this study, we integrate migration pattern variables into the traditional seismic fatality risk model to explain the concentration of earthquake fatalities in suburban areas. While ground motion intensity and population exposure account for higher seismic fatalities in suburban neighborhoods and urban–rural corridors during the Chi-Chi earthquake, the presence of migrants from low-income origins and a higher proportion of indigenous population further amplified the fatalities in these areas. It is important to note that our findings may not extend universally to all earthquakes, given the diverse geological and

tectonic structures, social contexts, institutional settings, and other uncertainties they involve. Nonetheless, low-cost housing options with a lack of safe infrastructure will disproportionally deteriorate fatality during earthquakes. This illustrates a pressing necessity to enforce building code regulations especially providing a safe housing and living standard for migrant workers without compromising affordability.

While the investigation of a single earthquake cannot be generally applied, it sheds lights on pathways for improving methods, assumptions, and implications critical for advancing seismic risk assessment studies and potentially impacting how extreme weather events and other catastrophes' fatalities are studied. Our empirical findings underscore the need to reexamine city-, region-, and nation-wide earthquake preparedness and response policies to better allocate resources to areas at greatest risk. These findings suggest that focusing seismic risk mitigation efforts in newly developed, economically disadvantaged suburban

neighborhoods and hub cities that cater rural-urban migrations, characterized by high concentrations of migrants living in informal housing or low-cost condominiums is important. Recognizing the suburban syndrome and implementing safety and affordability regulations in suburban housing could play a pivotal role in safeguarding human lives against future catastrophic earthquakes.

**Code availability**

The code for migration pattern and statistical analyses are available on GitHub (https://github.com/karenthchen/Migration-seismic-risk). Maps presented in this paper are generated using QGIS, a freely available software.

**Author contribution**

All authors designed research; T-H.K.C. performed research; T-H.K.C, G-Y.L., and C-H.Y. curated data; and T-H.K.C, K-
H.E.L, T.H.L, and D.M.C wrote the paper.

**Competing interests**

The contact authors have declared that none of the authors has any competing interests.



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
