# Peer review of "Migration as a Hidden Risk Factor in Seismic Fatality: A Spatial Modeling of the Chi-Chi Earthquake and Suburban Syndrome"

_EGUsphere, 2024_

## Author Response (AR1)

**Response to reviewers' comments and recommendations**

Ref: Ms. No. egusphere-2024-1493

Title: Migration as a Hidden Risk Factor in Seismic Fatality: A Spatial Modeling of the Chi-Chi Earthquake and Suburban Syndrome

We sincerely thank the reviewers for all the constructive comments. Following the suggestions and comments, in the revised version, we have included additional analysis, clarified our methods, and carefully revised the manuscript. We provide a point-by-point response to each comment:

**Reviewer #1**

| Comment | Response |
|---|---|
| The study presented in this paper explores the impact of migration on seismic fatality risk and introduces an interesting approach to seismic fatality risk modeling through the employment of a radiation model that accounts for migration patterns. The following comments are intended to provide suggestions to increase the impact of the paper and improve its clarity for the reader. | |
| 1. Considering the authors' selection of Sa03 as the intensity measure, is it correct to assume that the study area mainly consists of low-to-midrise buildings? If available, can the authors provide further information regarding the building typologies in the studied area, perhaps in terms of construction material and building height? | The Chi-Chi earthquake damaged 105,479 buildings, including 148 condominiums. This means that low-to-midrise buildings accounted for over 99.8% of the damaged structures (Ministry of Interior, 2017). We have added information about the building height and construction materials of the damaged buildings in the study area and method sections. *We selected Sa03 as the indicator of seismic intensity because it captures the response of low-to-midrise* |

*buildings (one to six stories), which comprised 99.8% of the buildings damaged during the Chi-Chi earthquake (Ministry of Interior, 2017).*

*The top three damaged structural types were reinforced concrete structures (44%), unreinforced brick structures (22%), and unreinforced clay block buildings (12%) (Tsai et al., 2000).*

2.  Do the authors think that the daily time-based fluctuation could be incorporated into their model? For example, higher occupancy in residential buildings during nighttime compared to daytime could impact the results (s. FEMA P-58/BD-3.7.8 Casualty Consequence Function and Building Population Model Development). Please comment on this possibility.

Incorporating daily time-based fluctuations into the model would indeed help to better understand the role of mobility in seismic fatalities. However, capturing this pattern for our case, which occurred in 1999, is challenging. The method proposed by FEMA, which relies on land use types, may not be suitable for our study area, as many commercial land uses are mixed with residential ones (e.g., shops on the ground floor with residential units above), and the proportion of this mix is uncertain. Currently, day- and night-time population data for Taiwan is only available from the period after mobile phone technology became widespread (e.g., Liu and Huang 2020).

We discussed this limitation the discussion section:
*First, given the early timeline of the case study (1999), we were unable to validate the migration patterns using newer technology. Previous studies that validated the model with empirical data have shown that radiation models predict commuting patterns well at the national level (Masucci et al., 2013; Simini et al., 2012), though they*

|  | *may underestimate long-distance and international migration (Kluge and Schewe, 2021). Future studies on recent earthquakes could employ mobile data to further distinguish between different migration behaviors (e.g., daily commuting versus seasonal migration), which would allow for disaggregating the sources of seismic risks associated with mobility.* |

3.    The authors define the incidence rate ratio as the factor by which the fatality rate multiplies when that variable increases by one unit, assuming all other variables remain constant. For some of the variables, it is not clear how this increase is calculated. For instance, assuming the fault-impacted area is 0.02, does an increase by one unit mean the area becomes 0.03? Please provide details.

The effect of the IRR is based on the absolute unit of the variable. For the variable fault-impacted area, which ranges from 0 to 1, one unit represents 100% coverage. Since most neighborhoods are only partially overlapped with the fault-impacted area, we interpret the IRR in increments of 0.1 (or 10%) for practical understanding.

The IRR for fault-impacted area is 10, meaning that an increase in fault zone coverage from 0% to 100% will lead to a 900% increase in fatalities. Therefore, a 10% increase in fault zone coverage (i.e., 0.1 or 100%/10) results in a 90% increase in fatalities (i.e., 900%/10).

We added an example of the interpretation of IRR in our method section.

> *The incidence rate ratio (IRR) of a given variable can be interpreted as the factor by which the fatality rate multiplies when that variable increases by one unit, assuming all other variables remain constant (Clayton and Hills, 2013). For example, if the IRR for the variable fault ratio is 2.65,*

*it means that an increase in the fault ratio from 0% to 100% will lead to a 165% increase in fatalities (calculated as (2.65−1)×100%). In other words, for each 10% increase in the fault ratio, fatalities are expected to increase by 16.5%. When a variable is in logarithmic form, the IRR can be interpreted as the factor by which the fatality rate multiplies for a 1% increase in that variable.*

4.    Further, the reviewer questions the rationale behind the "increase by one unit" approach, as many variables have different units. It might be more effective to increase these variables by the same percentage rather than by one unit. For example, regarding spectral acceleration Sa03, increasing it by 1g would likely cause many buildings to collapse.

Consider the following scenario: On site A, assume a low-rise building with a spectral acceleration Sa03 at collapse of 0.6g (capacity). This means any Sa03 larger than 0.6g would cause the building to collapse (ignoring record-to-record variability). If the observed Sa03 on site A is 0.1g and we increase it by one unit, making it 1.1g, the expected fatalities due to collapse should not differ between 0.6g and 1.1g, as the fatality rate saturates at collapse, which occurs at 0.6g. Therefore, it may be more beneficial to increase the variables by the same percentage rather than by one unit for ease of comparison. Please comment.

When applying the percentage approach (elasticity), the variable is transformed into logarithmic form, which means that the effect of increasing the variable decreases as the baseline value increases. This can be useful for interpreting Sa03, particularly because its effect on fatalities may saturate at higher levels. However, this approach may not be suitable for other variables.

For example, with the proportion of the population above age 64, it is hard to assume that an increase from 10% to 20% has a larger effect than an increase from 80% to 90%. Additionally, it may be challenging to compare our results with baseline references that did not use this transformation.

To address this, we provide an additional model in the appendix where all variables are in log form, but we maintain the original model in the main text. In the appendix model, the IRR can be

| | interpreted as the effect of a one percent increase in the variable. We have also corrected the interpretation of log-transformed variables in the existing model, such as income, to reflect the effect of a percentage increase. |
|---|---|
| 5. Does the number of fatalities saturate at the spectral acceleration level associated with collapse? If no, it is suggested to mention this as a limitation of the study. | In our study, the number of fatalities saturated at Sa03 ~= 1.0g (Fig. S1), meaning that an additional increase in the spectral acceleration may not cause more fatality. We have added this limitation to our discussion.
[Figure]
 Figure S1. Distribution of fatalities to Sa03(g). *We did not account for the saturation effect of Sa03 — that is, the possibility that higher Sa03 values might not cause additional damage beyond a certain threshold.* |
| 6. The seismic hazard clearly has a strong influence on the number of fatalities. However, regarding the incidence rate ratios in Table 2, the high ratio associated with Sa03 may be due to the effect discussed in Comment 4. As a result, a direct comparison between variables based solely on their incidence rate ratios (computed by the "increase by one unit" approach) may be challenging. | As our response to comment point 4, we added a log-form model to the appendix. |

**Reviewer #2**

| Comment | Response |
|---|---|
| This paper investigates the 1999 Chi-Chi earthquake in Taiwan across 4,052 neighborhoods, employing Poisson regression and maximum likelihood estimation to predict incidence rate ratios and determine the significance of various covariates related to hazard, exposure, vulnerability, and migration patterns. The authors used the radiation model to estimate migration patterns and examine their effect on seismic risks and fatalities. The topic is highly relevant, the study is well-constructed, and the methodology is both innovative and thoroughly explained. However, I have a few comments for the authors to consider: | Thank you for your encouraging words and constructive comments. |
| Introduction:
Line 31: Replace "natural disaster" with "natural hazard" for accuracy. | We have corrected this. |
| Line 139: Consider replacing "resource scarcity (vulnerability)" with just "vulnerability" to avoid conflating different concepts, as vulnerability refers to the potential for loss, whereas resource scarcity is a distinct term. | We have replaced "resource scarcity (vulnerability)" with just "vulnerability". |
| Methods:
Line 158: Update to "In this study, we adopted the neighborhood geographic unit to estimate the socio-spatial effects of …" to improve readability and relevance. Additionally, I would suggest incorporating the risk formula into either the Data or Models section for clarity. | We have revised the sentence accordingly. We also added the risk formula in the Data section because it guides the structure of our data. |
| Consider adding a map to illustrate your study area at the neighborhood level. You could include a boundary in Figure 2 to enhance understanding. | We have included the neighborhood boundary in Figure 2. |
| Line 180: Add "n=4502" in Table 1 for clarity. | Added to Table 1's caption. |
| Reorganize section "2.2.4 on Independent | We have reorganized this section to follow |

| | |
|---|---|
| Variables: Vulnerability" to follow the order presented in Table 1. For example, in Line 214, specify that "data was collected from…" and again in Line 216 for the household income part. Merge the "proportion of indigenous population" information with Line 236 to improve readability and flow. | the order in Table1, starting from demographic variables, followed by income variables, and finally indigenous population variables. We also moved the data source to right after the brief intro of each variable. |
| Line 259: Equation 1 does not include "rij" as indicated in lines 258-261. | We have revised the description to make sure all symbols are presented in the equation. |
| For Equations 2 and 3, please include the left-hand side of the equations for completeness. | We have added the symbols at the left side of equations 2 and 3. |
| Models: Some variables have positive effects (e.g., high income), while others have negative effects (e.g., income disparity). Consider adding a more detailed explanation of how these effects are demonstrated within your models. | We have added our hypothesis regarding positive and negative effects in models. *While we hypothesized that most variables would have a positive effect on fatality risk, some may have negative effects. For instance, median household income is associated with greater resources to cope with earthquakes. Similarly, although older groups may be more vulnerable due to physical limitations, they may also possess more experience in dealing with earthquake situations.* |
| Regarding the migration pattern calculated using the radiation model, it would be beneficial to discuss the model's precision and accuracy. This will help clarify the model's strengths and limitations in predicting real-world migration flows and identifying areas where the model may over- or underestimate movements. Such a discussion could guide future refinements or improvements in its application. | We have added a discussion of the model's precision and accuracy based on the literature. |
| Results:
 1. Page 14: Please specify the exact number of fatalities in suburban or urban fringe areas. | The 1999 Chi-Chi earthquake resulted in 2,444 deaths, including 1,049 located in suburbs and the urban-rural fringe. We have included the exact number to section 3.2. |

| | | |
|---|---|---|
| 2. | Include the boundaries of the study areas in Figure 3. The text on vulnerability in Figure 3 is confusing. | We added a legend of study area in Figure 3. We also added a bracket to indicates where the text of vulnerability refers to. |
| 3. | Lines 355-356: The discrepancy in the effects of population over age 64 could be due to higher resolution data, but another potential explanation is that older adults may have more experience and unique perceptions in earthquake situations, which could influence the results. Consider discussing this in more depth. | We discussed additional possible explanation of the age 64 variable. *Similarly, although older groups may be more vulnerable due to physical limitations, they may also possess more experience in dealing with earthquake situations.* |
| 4. | Ensure consistency throughout the paper when reporting significance (e.g., use "P" versus "p"). | We have consistently corrected the significance terms. |